# The Effects of Cathepsin B Inhibition in the Face of Diffuse Traumatic Brain Injury and Secondary Intracranial Pressure Elevation

**DOI:** 10.3390/biomedicines12071612

**Published:** 2024-07-19

**Authors:** Martina Hernandez, Sean Regan, Rana Ansari, Amanda Logan-Wesley, Radina Lilova, Chelsea Levi, Karen Gorse, Audrey Lafrenaye

**Affiliations:** Department of Anatomy and Neurobiology, Virginia Commonwealth University, Richmond, VA 23298-0709, USA; martina.hernandez@duke.edu (M.H.);

**Keywords:** cathepsin B, intracranial pressure, neuronal membrane disruption, rat, somatosensory sensitivity, traumatic brain injury

## Abstract

Traumatic brain injury (TBI) affects millions of people each year. Previous studies using the central fluid percussion injury (CFPI) model in adult male rats indicated that elevated intracranial pressure (ICP) was associated with long-term effects, including neuronal cell loss and increased sensory sensitivity post-injury and secondary ICP elevation, which were not seen following injury alone. Investigations also indicated that cathepsin B (Cath B), a lysosomal cysteine protease, may play a role in the pathological progression of neuronal membrane disruption; however, the specific impact of Cath B inhibition following CFPI remained unknown. Thus, the focus of this study was to evaluate the effects of Cath B inhibition via the intracerebroventricular infusion of the Cath B inhibitor to the CA-074 methyl ester (CA-074Me) 2w following injury with or without secondary ICP elevation. This was accomplished using adult male rats continuously infused with CA-074Me or 10% DMSO as a vehicle control for 2w following either sham injury, CFPI only, or CFPI with subsequent ICP elevation to 20 mmHg. We assessed Cath B activity and evaluated the protein levels of Cath B and Cath B-binding partners AIF, Bcl-XL, and Bak. We also conducted histological analyses of the total cell counts to assess for cell loss, membrane disruption, and Cath B localization. Finally, we investigated somatosensory changes with the whisker nuisance task. Overall, this study demonstrated that Cath B is not a direct driver of membrane disruption; however, the administration of CA-074Me alters Cath B localization and reduces hypersensitivity, emphasizing Cath B as an important component in late secondary pathologies.

## 1. Introduction

Traumatic brain injury (TBI) affects approximately 60 million people globally [1] and 2.8 million people in the United States each year [2], with a rate of 759 TBIs for every 100,000 individuals reported globally in 2016 [3]. The annual number of TBIs is likely even higher as less severe TBIs often go unreported. This life-altering event can produce impairments in motor, cognitive, sensory, and affective function that can persist chronically [4], precipitating significant costs to patients, families, and communities. Moreover, secondary elevations in intracranial pressure (ICP) to levels at or above 20 mmHg following TBI can increase TBI-induced morbidity and mortality [5,6]. The current treatments and strategies for managing ICP elevation following a TBI are limited, with the most aggressive treatments involving invasive craniectomies [5]. While we have some indications of the pathological changes induced by elevated ICP following TBI, such changes, particularly those associated with diffuse TBI pathologies, are difficult to investigate in human patients and the molecular cascades involved remain nebulous. Therefore, we utilized the central fluid percussion injury (CFPI) model in rodents to replicate the diffuse pathologies induced by TBI and pair the injury with specific manipulations of ICP [7].

Previous studies from our lab using this CFPI model in adult male rats indicated that elevated ICP to 20 mmHg is associated with neuronal cell loss and increased sensory sensitivity weeks following CFPI and secondary ICP elevation, but these changes are not seen following injury alone [7]. Additionally, acute neuronal membrane disruption is nearly doubled with secondary ICP elevation hours following TBI [7,8]. Neuronal membrane disruption refers to the perturbation of the neuronal plasma membrane and is visualized using cell-impermeable tracers, such as dextran conjugated with fluorescent labels. Membrane disruption has been demonstrated to occur acutely and sub-acutely both in vitro and in vivo in mice, rats, and swine following TBI and spinal cord injury [7,8,9,10,11,12,13,14,15,16,17,18,19]. Recently, it was found that neuronal membrane disruption in the lateral neocortex occurs biphasically, in which significant membrane disruption was seen sub-acutely (6 h–3 d post-CFPI) which is reduced at 1w post-CFPI and re-emerges at more chronic timepoints (2w–4w post-CFPI) [20].

Previous investigations indicated that cathepsin B (Cath B), a lysosomal cysteine protease, may play a role in the pathological progression of neuronal membrane disruption. Specifically, our group found that Cath B localizes out of the lysosomal compartment and into the cytosol, specifically in the membrane-disrupted neurons at 2w and 4w post-injury [8,21]. Cath B is known to participate in various cell damage and death processes [22,23,24,25,26,27,28,29,30,31,32,33]. Moreover, Cath B has been implicated in the pathologies seen in other models of TBI, such as protein upregulation, lysosomal permeability, cell death, and behavioral deficits [34,35,36].

Therefore, the focus of this study was to evaluate the impact of continuous Cath B inhibition for 2w following injury and ICP elevation. This was accomplished using adult male rats continuously treated with either the Cath B inhibitor CA-074 methyl ester (CA-074Me) or the vehicle for CA-074Me, 10% dimethylsulfoxide (DMSO) in sterile saline, for 2w following either a (1) sham injury, (2) CFPI, or (3) CFPI with subsequent ICP elevation to 20 mmHg. We found that Cath B did not appear to be a direct driver of membrane disruption at 2w following CFPI; however, Cath B may play a role in TBI-induced and ICP-exacerbated sensory hypersensitivity, warranting further investigations. We further found that 10% DMSO reliably reduced neuronal membrane disruption.

## 2. Materials and Methods

### 2.1. Animals

Experiments were conducted under protocol number AM10251 in accordance with the Virginia Commonwealth University institutional ethical guidelines concerning the care and use of laboratory animals (Institutional Animal Care and Use Committee, Virginia Commonwealth University), which adhere to regulations including, but not limited to, those set forth in the ARRIVE guidelines and in the Guide for the Care and Use of Laboratory Animals, 8th Edition (National Research Council). Animals were housed in individual cages on a 12 h light–dark cycle with free access to food and water. A total of 78 adult male Sprague Dawley rats weighing 320–420 g were used for this study (n = 36 for protein analyses, n = 36 for histological assessments). There were 12 animals in each group: (1) sham injury with 10% DMSO infusion, (2) sham injury with CA-074Me infusion, (3) TBI with 10% DMSO infusion, (4) TBI with CA-074 infusion, (5) TBI + ICP elevation with 10% DMSO infusion, and (6) TBI + ICP elevation with CA-074Me infusion (n = 6/group for histological assessments, n = 6/group for molecular assessments). Animals used for molecular and histological studies were equally distributed throughout the duration of the study to allow for verification of CA-074Me efficacy throughout. The full experimental design schematic is represented in Figure 1. Our a priori exclusion criteria included weight loss of more than 20% or gross brain pathology (contusion, subdural hematoma, or gross tissue loss). Six animals met exclusion criteria in this study and were excluded. All surgeries were conducted by the same surgeon during similar hours of the day to minimize potential variability. Animal injury state and drug infusions were randomly determined using a random number generator by a separate investigator from the surgeon.

### 2.2. Drug Preparation

The solid form of the Cath B inhibitor CA-074 methyl ester (CA-074Me; MedChemExpress, Cat#: HY-100350, Monmouth Junction, NJ, USA) was stored at −20 °C until reconstitution in 100% dimethylsulfoxide (DMSO) at a concentration of 100 µg/µL. The 31 µL reconstituted aliquots were stored at −80 °C. Prior to surgery, an investigator separate from the surgeon used a random number generator to determine the animal’s injury and inhibitor infusion group. For CA-074Me infusion animals, a 100 µg/µL CA-074Me/DMSO aliquot was thawed and diluted with sterile 0.9% saline to a final concentration of 10 µg/µL in 10% DMSO by an investigator separate from the surgeon. This was carried out to avoid precipitation of CA-074Me, which occurred if the compound was frozen with any amount of 0.9% saline or if the percentage of DMSO was lower than a final concentration of 10% at either room temperature or 37 °C. The resulting solution (CA-074Me in 10% DMSO-sterile saline or 10% DMSO-sterile saline) was pipetted into tubes labeled with the date of the procedure to maintain surgeon blinding and stored at 37 °C. Connection of the brain canulation kit (Alzet; Cat. # 0004760; Cupertino, CA, USA) to the 250 µL mini osmotic pump (Alzet; Model 2002), and the subsequent filling with blinded solution was performed by the surgeon the day prior to surgery following. The prepared pump with blinded solution was placed in a sterile tube with saline to initiate osmotic pressure and then incubated overnight at 37 °C alongside the tube containing the bolus infusion.

### 2.3. Whisker Nuisance Task

Hypersensitivity to whisker stimulation in awake-behaving rats was assessed using the whisker nuisance task (WNT) as has been carried out previously [7,37,38]. Briefly, animals were assessed prior to injury and at 13 d post-sham or CFPI by an investigator blind to the animal group (Figure 1). For this assessment, the animal was placed inside a clean plastic open field lined with a chux pad and allowed to acclimate for 5 min. A wooden applicator stick was brushed along both sides of the rat’s whiskers for a 5 min period followed by 1 min rest. This sequence was repeated three total times. During these trials, the following behaviors were scored on a scale of 0 to 2 (0 = normal, 1 = exhibits some nuisance behavior, 2 = exhibits profound nuisance behavior): (1) movement, (2) stance and body position, (3) breathing, (4) whisker position, (5) whisker response, (6) evading stimulation, (7) response to stick presentation, (8) grooming, (9) ear position, (10) sniffing, (11) fur ruffling, and (12) urination/defecation (Appendix A). The individual behavior scores for each animal were summed for each trial with the highest possible score being 24, and the sums were averaged for the three trials. A higher WNT score indicates more pronounced agitation/sensitivity.

### 2.4. Surgical Preparation, Injury Induction, and Drug Infusion

On the day of injury, animals were anesthetized with 4% isoflurane in 30% O_2_ and 70% room air and ventilated with 2% isoflurane in 30% O_2_ and 70% room air throughout the duration of the surgery, injury, and post-injury physiological monitoring. Heart rate, respiratory rate, and blood oxygenation were monitored via a hind-paw pulse oximetry sensor (STARR Life Sciences, Oakmont, PA, USA) for the duration of anesthesia, except during the induction of injury. Body temperature was maintained at 37 °C using a rectal thermometer connected to a feedback-controlled heating pad (Harvard Apparatus, Holliston, MA, USA). All animals were placed in a stereotaxic frame (David Kopf Instruments, Tujunga, CA, USA). A midline incision was made, followed by a 4.8 mm circular craniectomy along the sagittal suture midway between bregma and lambda for placement of the injury hub. Additionally, a 2 mm burr hole was drilled in the left parietal bone overlying the left lateral ventricle-located 0.8 mm posterior, 1.3 mm lateral, and 2.5–3 mm ventral to bregma. Through this burr hole, a 25-gauge needle connected to a pressure transducer and micro-infusion pump (11 Elite syringe pump; Harvard Apparatus) via sterile saline-filled PE50 tubing was placed into the left lateral ventricle. Appropriate placement was verified via a 1.3 μL/min infusion of sterile saline within the closed fluid-pressure system during needle placement, with a drop in pressure indicating breach of the lateral ventricle [8,10]. This method of validating ventricle cannulation does not increase the ICP. The needle was held in the ventricle for at least 5 min while recording pre-injury ICP. After this pre-injury reading, the needle was slowly removed, and the burr hole was covered with bone wax before preparation for sham or CFPI [7,39]. Briefly, a Luer Lock syringe hub was affixed to the craniectomy site, and dental acrylic (methyl-methacrylate; Hygenic Corp., Akron, OH, USA) was applied around the hub and allowed to harden. Following hardening of the dental acrylic, anesthetized animals were removed from the stereotaxic frame and injured with a fluid pulse of 2.05 ± 0.16 atmospheres lasting ~22 ms (Figure 1). The pressure pulse was measured by a transducer affixed to the injury device and displayed on an oscilloscope (Tektronix, Beaverton, OR, USA). Identical surgical procedures were followed for sham-injured animals, without release of the pendulum to induce injury [7,20]. Immediately after the injury, the animal was reconnected to the ventilator and physiologic monitoring device, and the hub and dental acrylic were removed en bloc. Surgifoam was placed over the craniectomy/injury site. The animal was then replaced in the stereotaxic device and the ICP probe was reinserted into the lateral ventricle, as described above, for post-injury ICP monitoring. At 15 min following injury, animals were subcutaneously administered 0.9 mg/kg buprenorphine-HCl in a slow release as an analgesic and then monitored for 1 h following injury (Figure 1). Monitoring of ICP was carried out for both sham and TBI-only animals. For animals in the TBI and 20 mmHg-ICP elevation group, at 15 min post-CFPI, ICP was elevated to 20 mmHg via infusion of sterile normal saline at 1.3–13 µL/min using the micro-infusion Pump 11 Elite syringe pump controlled by the experimenter. Once 20 mmHg was achieved, ICP was maintained at 20 mmHg for 1 h (Figure 1).

One hour following injury and at the conclusion of the ICP monitoring or elevation, a 12.5 µL bolus of either 10 µg/µL CA-074Me or 10% DMSO was infused into the left lateral ventricle. Following bolus infusion, the caudal end of the midline incision was blunt dissected to lift the skin over the shoulder blades, allowing for the implantation of the osmotic pump and brain infusion cannula. The brain infusion cannula was placed into the burr hole for access to the left lateral ventricle using a stereotaxic cannula holder located 0.8 mm posterior, 1.3 mm lateral, and 2.5–3 mm ventral to bregma. The cannula was held in place with cyanoacrylate. The incision sites were then sutured and treated with lidocaine and triple-antibiotic ointment, after which the animals were recovered and returned to a clean home cage (Figure 1). The mini osmotic pump infused at a rate of 0.5 µL/h for the entire 2w post-injury period.

### 2.5. Tissue Processing for Molecular Analysis

Following the 2w WNT assessment, animals to be used for molecular studies (n = 6/group; n = 36 total) were injected with 150 mg/kg euthanasia-III solution (Henry Schein, Melville, NY, USA) and underwent transcardial perfusion with cold 0.9% saline. In these animals, both left and right lateral neocortices were dissected and frozen at −80 °C for protein expression and activity assessments.

### 2.6. Tracer Infusion and Tissue Processing for Histological Analysis

In the animals being prepared for histological assessment (n = 6/group; n = 36 total), fluorescently tagged dextran (0.6 mg/25 µg in sterile 0.9% saline; ~1.6 mg/kg) was infused into both lateral ventricles 1 h prior to sacrifice (2w post-sham or CFPI), as described previously [8,20]. Briefly, the burr hole overlying the left lateral ventricle was reopened and another burr hole was drilled over the right lateral ventricle located 0.8 mm posterior, 1.3 mm lateral, and 2.5–3 mm ventral to bregma. In addition, 12.5 µL of 10 kDa dextran conjugated to Alexa-Fluor-488 (Invitrogen, Thermo Fisher Scientific, Cat#: D22910, Waltham, MA, USA) were infused into the left then right lateral ventricle at 1.3 µL/min with continuous ICP monitoring. The tracer was allowed to diffuse throughout the parenchyma for 1 h prior to transcardial perfusion (Figure 1).

Then, 1 h following dextran infusion into the lateral ventricles, animals used for histological assessments underwent transcardial perfusion with cold 0.9% saline followed by a switch to 4% paraformaldehyde/0.2% glutaraldehyde in Millonig’s buffer (136 mM sodium phosphate monobasic/109 mM sodium hydroxide) to fix the brain for subsequent immunohistochemical processing and analysis (Figure 1). After transcardial perfusion, the brains were removed, post-fixed for 24–48 h and coronally sectioned at a thickness of 40 µm in 0.1 M phosphate buffer using a vibratome (Leica, Banockburn, IL, USA) from the level of bregma to ~4.0 mm posterior to bregma. Sections were collected serially in 12-well plates and stored in Millonig’s buffer at 4 °C. A random starting well (#1–12) was selected using a random number generator and four serial sections were used for histological analyses. All histological analyses were restricted to layers V and VI of the lateral somatosensory neocortex extending from the area lateral to CA1 to the area lateral to CA3 of the hippocampus.

### 2.7. Quantification of Cathepsin B Activity

Portions of the fresh dissected left and right lateral neocortices and livers of the sham, TBI, and TBI + ICP animals were mechanically homogenized in 50 μM citric acid at pH 6.0, spun at 12,000× *g* at 4 °C for 10 min, and the supernatant of the whole homogenate was collected. Protein concentrations were measured using a NanoDrop Lite (Thermo Fisher Scientific, Waltham, WA, USA) and Cath B activity was measured in a 96-well plate, each well containing 2× reaction assay buffer (100 mM sodium acetate pH 5.5, 2 mM EDTA, 200 mM sodium chloride, 8 mM DTT), 2 μg of neocortex whole homogenate, and 1 mM Z-Phe-Arg-7-amino-4-(trifluoromethyl) coumarin (ZFR-AMC), a substrate for cysteine proteases that fluoresces upon cleavage by Cath B [21,34,35,40]. The plate was read on a PHERAstar microplate reader (BMG Labtech, Cary, NC, USA) at 60 min post-substrate addition at 365/450 nm excitation/emission. Each sample was loaded in triplicate per plate and was run on three independent plates to reduce pipetting and run-to-run variability biasing the results. A positive control well with 5 ng of purified human liver Cath B and a negative control well with only assay buffer and substrate were included on every plate. The raw arbitrary fluorescent values depicting activity were measured and expressed as fold-increase relative to the negative control, which was designated as zero-fold activity.

### 2.8. Western Blotting

Portions of the fresh dissected left and right lateral neocortices were homogenized in Western lysis buffer (150 mM NaCl, 50 mM Tris, pH 8.0, 1% Triton^TM^ X-100) and protease inhibitor cocktail (AEBSF 10.4 mM, Aprotinin 8 μM, Bestatin 400 μM, E-64 140 μM, Leupeptin 8 μM, Pepstatin A 150 μM; Sigma, Cat#: P8340, Saint Louis, MO, USA). Protein concentrations were measured using a BCA Protein Assay (Pierce Biotechnology, Thermo Fisher Scientific, Cat#: 23227) according to manufacturer’s instructions and read on a PHERAstar plate reader (BMG-Lab Tech). Protein (20 μg for Cath B, 5 μg for Bcl-XL, 10 μg for Bak and AIF) was boiled for 10 min in 50 mM dithiothreitol (Bio-Rad, Cat#: 1610610, Hercules, CA, USA), 2× Laemmli loading buffer (Bio-Rad, Cat#: 1610737) and run at 200 volts for 30 min on Mini-PROTEAN TGX Stain-Free 4–20% precast polyacrylamide gels (Bio-Rad, Cat#: 4568096). Protein was transferred onto 0.45 μm PVDF membranes using a Bio-Rad Transblot Turbo transfer system set to the Mixed Molecular Weight manufacturer setting (1.3–2.5 amps, 25 volts for 7 min). Western blotting was completed on an iBind Flex apparatus (Invitrogen, Thermo Fisher Scientific) using primary antibodies rabbit anti-Cath B (Cell Signaling Technology, 1:1000; Cat#: 31718S, RRID:AB_2687580, Danvers, MA, USA), rabbit anti-Bak (Cell Signaling Technology, 1:1000; Cat#: 12105S, RRID:AB_2716685), rabbit anti-Bcl-XL (Cell Signaling Technology, 1:1000; Cat#: 2764S, RRID:AB_2228008), or rabbit anti-AIF (Cell Signaling Technology, 1:1000; Cat#: 5318S, RRID:AB_10634755). Secondary antibody anti-rabbit-HRP (Jackson ImmunoResearch Laboratories, 1:5000; Cat#: 111–035-003; RRID:AB_2313567, West Grove, PA, USA) was used for all assays. Total protein (Stain-Free) and chemiluminescent images were taken on a ChemiDoc imaging system (Bio-Rad) holding the exposure time consistent for each protein assessed. Densitometric analyses of Cath B, AIF, Bak, and Bcl-XL were performed in FIJI ImageJ version 1.54f (National Institutes of Health; Bethesda, MD, USA). Cath B, AIF, Bak, and Bcl-XL protein bands were normalized to total loaded protein then to a naive control sample that was included in all runs. Total protein was used for loading normalization in order to avoid potential issues with changes in any normalization protein following TBI. All Western blots were run in triplicates on three separate gels to reduce run-to-run variability, potentially biasing the results.

### 2.9. Membrane Disruption and Total Cell Count Analysis

Consistent with previous studies, we assessed the potential for neuronal membrane disruption via the utilization of 10 kDa dextran, which is impermeable to cells with intact membranes [8]. Fluorescently tagged dextran-containing cells, indicative of membrane perturbation, as well as dextran in the parenchyma was visible without immunohistochemistry. To identify individual neurons, four tissue sections processed for histological assessment from each animal (n = 6/group, n = 36 total) were stained with 1:500 dilution of NeuroTrace 435/455 blue fluorescent Nissl Stain (Life Technologies, Thermo Fisher Scientific, Cat# N21479). The tissue was mounted onto slides using Vectashield Vibrance mounting medium (Vector Laboratories, Cat# H-1700; Burlingame, CA, USA). Sections were imaged with fluorescent optical sectioning microscopy using a Keyence BZ-X800 microscope (Keyence Corporation of America, Itasca, IL, USA). Quantitative analysis was performed as described previously [7]. Briefly, 20–35 images/animal of the left neocortical region were taken at 40× magnification in a systematically random fashion by a blinded investigator using NeuroTrace to verify focus. As the intensity of the fluorescently tagged dextran decreases as it diffuses away from the lateral ventricle (from layer VI and layer V), image acquisition settings were held constant for comparable layers for all groups analyzed.

To assess for cell loss, the total number of NeuroTrace-stained neurons were counted using the Hybrid Cell Counter function of the Keyence BZ-X800 Analyzer software version 1.1.1.8. Briefly, images were set to simple thresholding with uniform brightness and no smoothing. Auto-thresholding was used to highlight the NeuroTrace-positive label followed by watershed cell separation and removal of cell bodies smaller than 15 μm in diameter to exclude smaller glial cells and fragments of cell bodies from the analysis.

The number of NeuroTrace+ neurons exhibiting dextran uptake, in which the cell-impermeable fluorescently tagged dextran was visible within the NeuroTrace+ cell, not only throughout the parenchyma around the cell, was counted by eye using the FIJI ImageJ cell counting plug-in. Dextran-containing neurons were quantified for each image and averaged for each animal.

### 2.10. Cellular Cathepsin B Localization Analysis

To visualize Cath B, three sections/animal that was processed for histological analysis (n = 36 total) were immunohistochemically labeled. Tissue sections were blocked with 5% normal goat serum (NGS), 2% bovine serum albumin (BSA), and permeabilized with 1.5% Triton^TM^ X-100 for 2 h at room temperature. This was followed by immunolabeling using primary antibodies rabbit anti-Cath B (Cell Signaling Technology; 1:700; Cat#: 31718; RRID:AB_2687580) and mouse anti-NeuN (MilliporeSigma; 1:500; Cat#: MAB377; RRID:AB_2298772, Temecula, CA, USA). Tissue was incubated in secondary antibodies Alexa-647 conjugated goat anti-mouse (Life Technologies, Thermo Fisher Scientific, 1:700; Cat#: A32728, RRID:AB_10563566) and Alexa-568 conjugated goat anti-rabbit (Life Technologies, Thermo Fisher Scientific, 1:800; Cat#: A11036; RRID:AB_2534102), and the tissue was mounted onto slides using Vectashield hardset mounting medium with 4′,6-diamidino-2-phenylindole (DAPI) (Vector Laboratories, Cat#: H-1500). Quantitative analysis was performed as described previously [8,20]. Briefly, an investigator blinded to the animal group took five to six images within the left lateral cortex from three sections/animal at 40× magnification using a Keyence BZ-X800 microscope by using the dextran tag to verify images from animals that had membrane-disrupted dextran-containing neurons included neurons with and without membrane disruption. Image acquisition settings were held constant for comparable regions (layer V or VI). The images, including 488-dextran, DAPI, and Cath B, were analyzed in FIJI ImageJ. DAPI was used to identify nuclei of cells containing and not containing 488-dextran. Ten non-disrupted and ten membrane-disrupted cells were assessed in each image. If a total of ten neurons that were membrane disrupted could not be found in any given image, an equal number of membrane disrupted and non-disrupted neurons were captured in that image; n = 6 animals/group (1) sham injury with 10% DMSO treatment, n = 373 cells, (2) sham injury with CA-074Me treatment, n = 424 cells, (3) TBI with 10% DMSO treatment, n = 442 cells, (4) TBI with CA-074Me treatment, n = 480 cells, (5) TBI + ICP elevation with 10% DMSO, n = 497 cells, and (6) TBI + ICP elevation with CA-074Me, n = 432 cells. Cells were classified by Cath B localization within puncta (intra-lysosomal) or diffusely distributed throughout the cell body (extra-lysosomal), as has been carried out previously [8,21]. The localization of Cath B was expressed as the percentage of the total membrane-disrupted or non-disrupted neuronal population assessed in each experimental group that demonstrated punctate (lysosomal) Cath B localization.

### 2.11. Statistics

Data were tested for normality prior to utilizing parametric or non-parametric assessments, which were conducted using SPSS software version 29 (IBM Corporation, Armonk, NY, USA). Animal numbers for each group were determined by an a priori power analysis using effect size and variability previously observed in the lab when assessing pathology between sham and injured groups using the CFPI model, an alpha of 0.05, and a power of 80%. One-, two-, or three-way analysis of variance (ANOVA) and Bonferroni post hoc tests were performed for all between-group analyses. Statistical significance was set to *p* < 0.05. Data are reported as mean ± standard error of the mean (S.E.M.).

## 3. Results

### 3.1. Cathepsin B Activity Was Decreased in the Left and Right Cortex after 2w of Continuous CA-074Me Infusion

To validate the effectiveness of the intracerebroventricular (ICV) administration of CA-074Me, we measured the activity of Cath B in both the left and right cortices of animals infused with either a 10% DMSO vehicle or CA-074Me into the left lateral ventricle over the 2w period following sham, CFPI, or CFPI with secondary ICP elevation to 20 mmHg. We also assessed the Cath B activity within the liver of each animal to verify that systemic Cath B activity was not significantly impacted with our ICV infusion paradigm. There was no difference in the Cath B activity in the liver between the injury groups’ sham, CFPI, CFPI + ICP elevation (one-way ANOVA injury group *F*_2, 93_ = 1.2, *p* = 0.3). As expected, Cath B activity in the brain was significantly decreased in animals following the ICV infusion of CA-074Me compared to the 10% DMSO-infused animals (one-way ANOVA infusion group *F*_1, 93_ = 24.5, *p* = 3.0 × 10^−6^; Figure 2). The liver demonstrated much higher Cath B activity compared to either the left or right lateral neocortices (one-way ANOVA region *F*_2, 93_ = 41.5, *p* = 1.3 × 10^−13^); however, there was no significant difference in the Cath B activity within the left cortex compared to the right cortex (*p* = 0.85). This was consistent for each injury group in which the Cath B activity in the left and right cortex was significantly lower compared to the liver (liver vs. left cortex *p* = 1.7 × 10^−12^; liver vs. right cortex *p* = 3.0 × 10^−10^; Figure 2, Appendix A).

There was no interaction found between the inhibitor infusion and the injury group (two-way ANOVA *F*_2, 93_ = 0.21, *p* = 0.81). However, there was a significant interaction between the infusion of CA-074Me and the region assessed, in which a decrease in the Cath B activity was seen in both the left and right cortices of animals infused with CA-074Me ICV, which was not reflected in the liver (two-way ANOVA *F*_2, 93_ = 5.64, *p* = 0.005). A trend toward an interaction between the injury group and region was observed (two-way ANOVA *F*_4, 93_ = 2.306, *p* = 0.064). There was no interaction of the injury group, inhibitor infusion, and region for Cath B activity (three-way ANOVA *F*_4, 93_ = 1.013, *p* = 0.405).

### 3.2. Protein Expression of Cathepsin B and Signaling Partners Bcl-XL, Bak, and AIF

The overall expression of Cath B within the left lateral neocortex was evaluated in all injury and infusion groups. Western blot analysis revealed that the total Cath B protein levels were consistent across all groups (*F*_5, 30_ = 0.64, *p* = 0.67; Figure 3, Appendix A). There was no effect of the injury group (one-way ANOVA injury group *F*_2, 30_ = 0.1, *p* = 0.9) or inhibitor infusion (one-way ANOVA infusion group *F*_1, 30_ = 2.3, *p* = 0.14) on the Cath B protein quantity. There was no interaction between the injury group and infusion of CA-074Me on the Cath B protein quantity (two-way ANOVA *F*_2, 30_ = 0.35, *p* = 0.71). As we observed two distinct bands in our Cath B Westerns, we also assessed the potential changes in each band. There was no significant difference across the groups for the upper Cath B band (one-way ANOVA *F*_5, 30_ = 0.41, *p* = 0.83). While the lower Cath B was not significantly increased in any specific individual group (one-way ANOVA *F*_5, 30_ = 1.16, *p* = 0.35), there was a significant effect of the infusion group (one-way ANOVA *F*_1, 30_ = 5.25, *p* = 0.029) in which CA-074Me-infused animals had a higher expression of the lower Cath B band (Figure 3E). There was no effect of the injury group (one-way ANOVA *F*_2, 30_ = 0.03, *p* = 0.97) or interaction between the injury and infusion groups (two-way ANOVA *F*_2, 30_ = 0.24, *p* = 0.79) on the lower Cath B band.

Cath B has been shown to participate in various cell damage pathways through downstream proteins such as AIF, Bcl-XL, and Bak [22,23,30,41]. While a previous study indicated that the protein expression of AIF, as well as the expression ratio of the antagonizing proteins Bcl-XL and Bak, were consistent from 6 h to 4w following CFPI [21], the potential impact of secondary ICP elevation and/or Cath B inhibition following CFPI on such downstream proteins remained unknown. Therefore, we assessed the expression levels of Bcl-XL, Bak, and AIF in the left and right lateral neocortices in all injury and infusion groups.

There were no significant differences in the levels of the pro-survival protein Bcl-XL across all groups (one-way ANOVA *F*_5, 30_ = 0.26, *p* = 0.93; Figure 4, Appendix A). There was no effect on the Bcl-XL protein quantity from the injury group (one-way ANOVA injury group *F*_2, 30_ = 0.25, *p* = 0.78) or inhibitor infusion (one-way ANOVA infusion group *F*_1, 30_ = 0.69, *p* = 0.41). There was also no interaction between the injury group and infusion group (two-way ANOVA *F*_2, 30_ = 0.06, *p* = 0.94; Figure 4). As Bcl-XL presented as a doublet, we also assessed the potential changes in both the upper and lower bands independently. There were no differences between groups in either the upper band (*F*_5, 30_ = 0.32, *p* = 0.89) or lower band (*F*_5, 30_ = 0.3, *p* = 0.91) of this doublet. Quantities of the anti-survival Bak protein were also consistent across groups (*F*_5, 30_ = 0.05, *p* = 0.99; Figure 5, Appendix A). The injury group (one-way ANOVA injury group *F*_2, 30_ = 0.015, *p* = 0.99) and inhibitor infusion (one-way ANOVA infusion group *F*_1, 30_ = 0.05, *p* = 0.83) did not influence the Bak protein levels. No interactions between the injury group or inhibitor infusion were found (two-way ANOVA *F*_1, 30_ = 0.09, *p* = 0.91).

There were two distinct bands in the AIF Western, so both were assessed. The protein levels of both bands of AIF were consistent across all groups (upper band *F*_5, 30_ = 0.13, *p* = 0.99; lower band *F*_5, 30_ = 1.24, *p* = 0.33; Figure 6, Appendix A). The AIF protein levels were not directly impacted by the inhibitor infusion (one-way ANOVA infusion group upper band *F*_1,30_ = 0.13, *p* = 0.72; lower band *F*_1, 30_ = 0.12, *p* = 0.73) or injury group (one-way ANOVA injury group upper band *F*_2, 30_ = 1.98, *p* = 0.82; lower band *F*_2, 30_ = 0.8, *p* = 0.46). Despite there being a slight decrease in the AIF protein levels in the CFPI + ICP elevation group infused with CA-074Me primarily driven by changes in the lower AIF band for this group, these differences were not significant and there was no interaction between the injury group and infusion (two-way ANOVA upper band AIF *F*_2, 30_ = 0.05, *p* = 0.95; lower band AIF *F*_2, 30_ = 2.2, *p* = 0.13; Figure 6).

### 3.3. Elevation of ICP and/or Infusion of CA-074Me Did Not Impact the Number of Neurons in the Lateral Neocortex Following CFPI

Fluorescent NeuroTrace 435 was used to identify the Nissl substance within cells of the left lateral neocortex in sham, CFPI, or CFPI + ICP elevated animals given either 10% DMSO or CA-074Me (Figure 7).

The number of NeuroTrace+ cells with cell bodies > 15 µm^2^, to enrich for neuronal profiles, in each image (0.098 mm^2^) was counted to assess the potential for cell loss over the 2w post-injury period. There was no indication of cell loss in layers V and VI of the lateral neocortex in any group (*F*_5, 30_ = 0.362, *p* = 0.87; Figure 8, Appendix A). There was no effect on the total cell count from the injury group (one-way ANOVA injury group *F*_2, 30_ = 0.63, *p* = 0.54) or from inhibitor infusion (one-way ANOVA infusion group *F*_1,30_ = 0.47, *p* = 0.49). There was no interaction between the injury group and infusion group (two-way ANOVA *F*_2, 30_ = 0.043, *p* = 0.96; Figure 8).

### 3.4. Neuronal Membrane Disruption Was Not Significantly Altered Following CFPI in the Presence of 10% DMSO or CA-074Me Regardless of ICP Elevation

To determine the impact of Cath B inhibition via CA-074Me ICV infusion and/or secondary elevations in ICP following CFPI at 2w post-injury, we infused tagged dextran ICV prior to sacrifice at 2w. Fluorescent Nissl NeuroTrace+ neurons (>15 μm in diameter) containing the normally cell-impermeable dextran were considered membrane-disrupted and were quantified throughout layers V and VI of the left lateral neocortex (Figure 7). Neuronal membrane disruption was consistent in all groups (*F*_5, 30_ = 0.93, *p* = 0.47; Figure 9, Appendix A). Sham animals demonstrated levels of neuronal membrane disruption consistent with previous studies [7,20,38] in which the sham injury group infused with 10% DMSO had 7.83% ± 2.16 neurons sustaining membrane disruption and the sham injury group infused with CA-074Me had 12.09% ± 3.91 neurons sustaining membrane disruption (Figure 7A,D and Figure 9). Rats sustaining CFPI demonstrated membrane disruption in a much lower percentage of neurons than anticipated based on previous findings [7,20,38] in which the CFPI group infused with 10% DMSO had 6.56% ± 1.13 neurons sustaining membrane disruption and the CFPI group infused with CA-074Me had 10.86% ± 3.08 neurons sustaining membrane disruption (Figure 7B,E, and Figure 9). At percentages similar to those in sham and CFPI groups, animals sustaining CFPI with secondary ICP elevation to 20 mmHg also demonstrated minimal neuronal membrane disruption, in which the TBI + ICP elevation group infused with 10% DMSO had 10.72% ± 2.60 neurons sustaining membrane disruption and the TBI + ICP elevation group infused with CA-074Me had 6.34% ± 1.28 neurons sustaining membrane disruption (Figure 7C,F and Figure 9). There were no significant effects of the injury group (one-way ANOVA injury group *F*_2, 30_ = 0.19, *p* = 0.83), inhibitor infusion (one-way ANOVA *F*_1, 30_ = 0.45, *p* = 0.51), or interactions between the injury group and inhibitor infusion (two-way ANOVA *F*_2, 30_ = 1.92, *p* = 0.16).

### 3.5. Cathepsin B Re-Localizes from Lysosomes to Cytosol in Disrupted Neurons at 2w Following Injury

Re-localization/redistribution of Cath B out of the lysosome and into the cytosol has previously been seen in neuronal membrane-disrupted populations both acutely and weeks following CFPI [8,21]. The longer-term effects of secondary ICP elevation and/or Cath B inhibition on Cath B localization, however, remain nebulous. To investigate the potential changes in the localization of Cath B within neuronal membrane-disrupted and non-disrupted neurons, an immunohistochemistry for Cath B paired with quantitative image analysis was carried out [8,21]. The localization of Cath B within the lysosomal puncta was assessed for membrane-disrupted and non-disrupted neurons in each group at 2w post-injury (Figure 10). As expected, there was an effect of membrane disruption on Cath B localization (one-way ANOVA membrane disruption state *F*_1, 2633_ = 152.06, *p* = 5.2 × 10^−34^) in which Cath B was localized primarily within lysosomal puncta in non-disrupted neurons compared to neurons sustaining membrane disruption in most groups, in which Cath B was diffusely distributed throughout the cytoplasm of the soma (one-way ANOVA *F*_11, 2632_ = 20.46, *p* = 2.96 × 10^−40^, Bonferroni post hoc non-disrupted vs. membrane disrupted; sham CA-074Me *p* = 2.6 × 10^−17^; TBI 10% DMSO *p* = 4.67 × 10^−4^; TBI CA-074Me *p* = 0.025; TBI + ICP elevation 10% DMSO *p* = 0.014; TBI + ICP elevation CA-074Me *p* = 3.0 × 10^−6^; Figure 11, Appendix A). The only exception was the sham-injured group infused with 10% DMSO, in which there was no difference in the Cath B localization between the non-disrupted and membrane-disrupted neuronal populations (*p* = 0.084).

There was also an effect of the injury group on Cath B localization (one-way ANOVA injury group *F*_2, 2633_ = 9.24, *p* = 1.00 × 10^−4^) in which animals sustaining CFPI only had significantly fewer cells displaying punctate Cath B localization than sham-injured animals (*p* = 0.025) or animals sustaining a CFPI and ICP elevation to 20 mmHg (*p* = 2.20 × 10^−5^). Specifically, non-disrupted neurons in animals infused with CA-074Me and sustaining a CFPI had significantly lower Cath B localization within puncta compared to sham animals infused with CA-074Me (*p* = 3.59 × 10^−7^). In addition, there were differences in the membrane-disrupted neuronal population between the CFPI and the CFPI + ICP elevated groups infused with 10% DMSO (*p* = 0.014) and in the non-disrupted neuronal populations between the TBI and TBI + ICP elevated groups infused with CA-074Me (*p* = 0.035).

There was a significant effect of inhibitor infusion (one-way ANOVA infusion group *F*_1, 2633_ = 5.50, *p* = 0.019) on Cath B localization and there was a significant interaction between the neuronal membrane disruption state and inhibitor infusion (two-way ANOVA *F*_1, 2633_ = 5.80, *p* = 0.016). There was also an interaction between the inhibitor infusion and injury group (two-way ANOVA *F*_2, 2633_ = 3.53, *p* = 0.03). There was also a significant interaction of the membrane-disruption state, injury group, and inhibitor infusion (three-way ANOVA *F*_2, 2633_ = 10.11, *p* = 4.20 × 10^−5^). Specifically, the non-disrupted neuronal population in the sham group infused with CA-074Me had a significantly higher punctate localization of Cath B compared to the sham group infused with 10% DMSO (*p* = 0.021). There was also a significant shift in the Cath B localization in the non-disrupted neuronal population in the TBI group infused with CA-074Me compared to the TBI group infused with 10% DMSO (*p* = 0.029).

### 3.6. Secondary ICP Elevation Exacerbates Somatosensory Sensitivity at 2w Post-CFPI, Which Is Ameliorated with CA-074Me Infusion

To investigate the potential changes in somatosensory sensitivity following CA-074Me infusion, the whisker nuisance task (WNT) [7,37,38] was performed in animals prior to injury and at 2w following sham, CFPI, or CFPI and secondary ICP elevation to 20 mmHg for all groups in which a higher score indicated higher somatosensory sensitivity to whisker stimulation. Pre-injury scores were comparable across all groups: sham injury with 10% DMSO (3.33 ± 0.43; n = 11), sham injury with CA-074Me (3.43 ± 0.45; n = 13), TBI with 10% DMSO (4.58 ± 0.66; n = 12), TBI with CA-074Me (4.36 ± 0.42; n = 13), TBI + ICP elevation with 10% DMSO (3.79 ± 0.37; n = 11), and TBI + ICP elevation with CA-074Me (3.68 ± 0.53; n = 12). There was an effect between the pre- and post-WNT score from the injury group (repeated measures ANOVA *F*_1, 68_ = 87.24, *p* = 8.2 × 10^−14^; Figure 12, Appendix A). While the sham injury group infused with CA-074Me maintained a low WNT score at 2w (5.17 ± 0.63), the 2w post-injury WNT scores for animals in all other groups were significantly higher than the pre-injury scores (paired *t*-test; sham injury with 10% DMSO *p* = 0.009; TBI with 10% DMSO *p* = 0.02; TBI with CA-074Me *p* = 0.004; TBI + ICP elevation with 10% DMSO *p* = 8.5 × 10^−8^; TBI + ICP elevation with CA-074Me *p* = 0.004; Figure 12, Appendix A). There was a significant effect of the injury group (repeated measures ANOVA time and injury group *F*_2, 68_ = 8.01, *p* = 7.5 × 10^−4^) driven primarily by increases in the WNT score at 2w in the injured groups compared to the shams (sham vs. TBI *p* = 5.1 × 10^−4^; sham vs. TBI + ICP elevation *p* = 0.001). There was an effect of inhibitor infusion on the WNT score (two-way repeated measures ANOVA *F*_1, 68_ = 6.66, *p* = 0.012), in which groups given CA-074Me had lower WNT scores at 2w post-injury compared to those given 10% DMSO. There was no interaction of the time, inhibitor infusion, and injury group (three-way repeated measures ANOVA *F*_2, 66_ = 1.96, *p* = 0.148).

## 4. Discussion

Overall, the findings from this study show that CA-074Me, when given as a bolus followed by continuous osmotic pump infusion into the left ventricle, successfully inhibits Cath B activity in the left and right lateral neocortex (Figure 2). The Cath B activity was not significantly different between injury categories. The protein quantifications for Bak, Bcl-XL, and AIF also revealed no changes regarding CA-074Me treatment (Figure 4, Figure 5 and Figure 6). However, there was a small but significant increase in the protein levels of the lower band of Cath B in the groups given CA-074Me (Figure 3). This could indicate a compensatory mechanism in which cells express increased levels of Cath B in the face of reduced Cath B activity; however, no transcriptomic assessments to investigate the levels of Cath B mRNA were carried out. Therefore, further investigations regarding the expression and degradation of this lower Cath B band would be needed to rigorously investigate this possibility. Our histological assessments uncovered no cell loss in the lateral neocortex in layers V and VI in any group (Figure 8). We also did not observe a difference in the percentage of neuronal membrane disruption in any group regardless of the injury or inhibitor infusion (Figure 9). The assessments of Cath B localization recapitulated the reduced punctate Cath B localization in membrane-disrupted neurons compared to non-disrupted neurons (Figure 11), as has been shown previously [21]. The inhibition of Cath B also significantly altered Cath B localization (Figure 11). Somatosensory hypersensitivity, which was increased with injury and injury with elevated ICP, appeared to be reduced by CA-074Me infusion in animals sustaining CFPI with secondary ICP elevation (Figure 12).

The neuronal membrane disruption findings in the current study, particularly in the 10% DMSO vehicle control groups, were unanticipated based on previous studies. We previously found that while sham injury precipitated very little neuronal membrane disruption, approximately 20% of neurons were membrane-disrupted at 6 h following CFPI in adult male Fisher rats [8]. The percentage of neurons sustaining membrane disruption doubled to 40% in rats with naturally high ICP above 20 mmHg following CFPI [8]. A follow-up study using Sprague Dawley rats found similar percentages of neuronal membrane disruption, in which injured rats had ~25% membrane-disrupted neurons and injury with ICP elevation to 20 mmHg demonstrated ~62% membrane-disrupted neurons [7]. A study investigating the time course of membrane disruption from 6 h to 4w following a CFPI in male Sprague Dawley rats also found significantly higher percentages of neuronal membrane disruption following injury both acutely and weeks following injury [20]. Therefore, it was surprising to see no increase in the neuronal membrane disruption following CFPI and no exacerbation with secondary ICP elevation (Figure 8). However, while DMSO is mostly noted for its potentially toxic effects if used in high concentrations [42,43,44], DMSO has also been shown to have neuroprotective effects. In a study conducted by DiGiorgio and colleagues, using the lateral fluid percussion model (LFPI) in rats, the drugs alpha-tocopherol and curcumin were tested, with DMSO as the vehicle control and saline as a DMSO control [45]. In their assessments of cell damage/death, they found a significant reduction in Fluoro-Jade staining in all drug-treated groups suspended in a DMSO vehicle, including the DMSO-only group, which was not observed in the saline LFPI controls [45]. Earlier preclinical studies saw that the intravenous administration of this amphipathic molecule precipitated a reduction in the ICP [46,47]. DMSO was also assessed in the clinic as a potential treatment for intractable ICP elevation and was shown to rapidly reduce ICP [42,43,44,48]. However, it was challenging to administer clinically as the DMSO tended to break down IV tubing. In addition, DMSO could lead to hemolysis and hypernatremia/hyperosmolarity, which counteracted any positive impacts of lowering the ICP [42,43,44].

Biophysical assessments of DMSO have demonstrated increased permeability/flexibility in membrane dynamics [49,50]. DMSO’s effects on membranes were initially interrogated using molecular simulations in which Notman et al. found increased membrane flexibility, which was postulated to be linked to the formation of water pores [50]. Another group arrived at similar conclusions using atomic simulations [49]. These simulations were eventually tested in DC-3F Chinese hamster lung fibroblast cells exposed to different percentages of DMSO [51]. Membrane undulations, but not membrane permeability, were documented with low doses (<10%) of DMSO [51]. With DMSO concentrations closer to 10–20%, small membrane blebs were seen, in which Ca^2+^ diffused into the cells; however, there was no increased diffusion of the larger (630 Da) cell-impermeable Oxazole Yellow (Yo-Pro-1) molecule for this concentration of DMSO [51]. The dextrans used in the current study are 10 kDa, significantly larger than the Yo-Pro-1 used in the study conducted by Ménorval. Increased Yo-Pro-1 diffusion into cells was not significantly higher until ≥25% DMSO was administered to the cells [51]. Indeed, the addition of 5% DMSO to transected guinea pig spinal cord axons reduced the permeability of axonal membranes to a cell-impermeable tracer, even under conditions that the group had previously determined to significantly hinder membrane resealing [52]. In the current study, it may be possible that DMSO altered the membrane flexibility, thereby promoting membrane resealing by 2w post-injury, which could explain the lower amounts of membrane disruption we observed compared to previous studies [7,20,38].

We did, however, observe whisker sensory sensitivity increases following CFPI that was exacerbated by ICP elevations to 20 mmHg in our 10% DMSO-infused group, consistent with previously published studies [7,38]. Notably, CA-074Me administration in animals sustaining TBI and secondary ICP elevation lowered sensory hypersensitivity to levels consistent with the TBI-only groups infused with either 10% DMSO or CA-074Me (Figure 12). Previous groups found similar rescues in deficits with Cath B inhibition. Mice administered with CA-074 prior to a focal TBI showed the recovery of motor function and spatial learning compared to vehicle-treated animals [36]. Another study in which Cath B was knocked out of mice sustaining a focal TBI, or inhibited via the administration of E64d, a cysteine protease inhibitor, observed better motor behavior in animals with absent or inhibited Cath B compared to controls [35]. Based on these studies and our current finding, it is possible that Cath B inhibition may play a role in the exacerbation of whisker hypersensitivity following secondary ICP elevation; however, additional behavioral assessments would be needed to more fully explore this possibility. Overall, while the mechanisms behind membrane disruption are not directly coupled with Cath B, Cath B seems to have a role in the exacerbation of behavioral morbidities following TBI and secondary ICP elevation, offering a molecular target in treating the after-effects of elevated ICP following injury. Therefore, further study into this possibility is warranted.

## Figures and Tables

**Figure 1 biomedicines-12-01612-f001:**
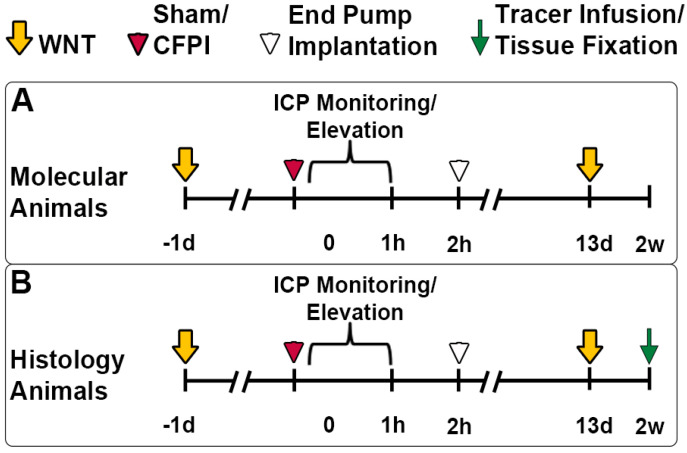
Schematic representation of the experimental procedures utilized in the current study. (**A**) The top panel represents the experimental design for animals processed for molecular studies and (**B**) the lower panel depicts the experimental design for animals processed for histological studies. All animals in both groups had whisker nuisance task (WNT, yellow arrow) behavioral assessments carried out prior to injury and at 2w (13 d) post-injury. On the day of injury, day 0, animals either sustained a central fluid percussion injury (CFPI) or a sham injury (red arrowhead). From 15 min to 1 h following sham or CFPI, animals either had intracranial pressure (ICP) monitoring or ICP elevation to 20 mmHg. An initial bolus of either the cathepsin B inhibitor CA-074Me or 10% DMSO vehicle followed by implantation of an osmotic pump for continuous intracerebroventricular (ICV) administration was completed 1 h to 2 h post-injury followed by recovery from anesthesia. At 2w post-sham or CFPI, animals used for histological analysis were anaesthetized for bilateral ICV infusion of the cell-impermeable 10 kDa Alexa-Fluor-488 fluorescently tagged dextran (green arrow), followed by transcardial perfusion 1 h later.

**Figure 2 biomedicines-12-01612-f002:**
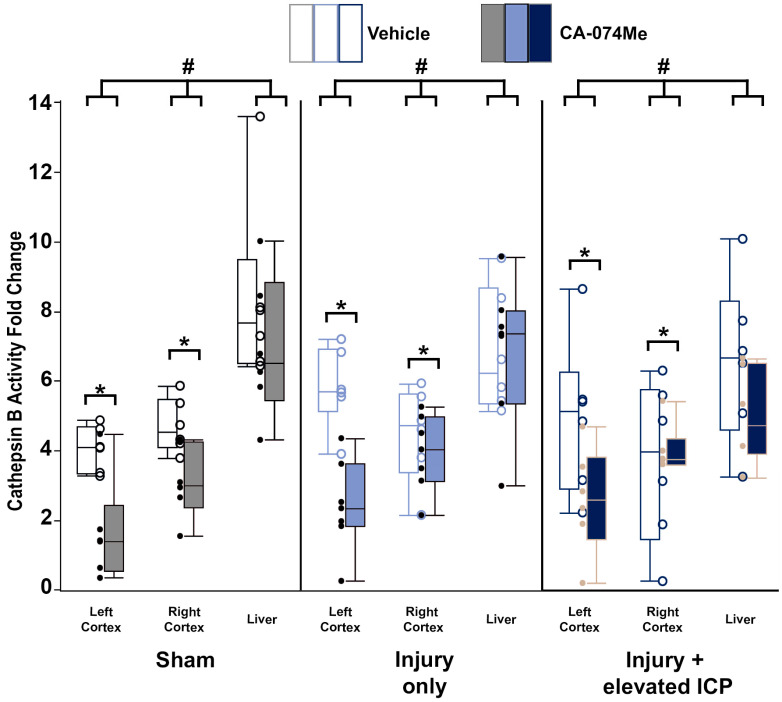
Cathepsin B (Cath B) activity was reduced following 2w of the Cath B inhibitor CA-074Me infusion into the left lateral ventricle. Cath B activity was significantly decreased in the left and right lateral neocortex following CA-074Me infusion (filled boxes) compared to 10% DMSO control (vehicle; unfilled boxes). Box and whisker graph depicting the average fluorescent intensity indicating Cath B activity in the right and left side of the lateral neocortex and the liver of sham-injured control animals (n = 12; grey unfilled boxes for n = 6 saline-treated animals and filled boxes for n = 6 CA-074Me-treated animals), animals sustaining a traumatic brain injury (TBI) only (n = 13; light blue unfilled boxes for n = 6 saline-treated animals and filled boxes for n = 7 CA-074Me-treated animals), or animals sustaining a TBI followed by secondary intracranial pressure (ICP) elevation (n = 12; dark blue unfilled boxes for n = 6 saline-treated animals and filled boxes for n = 6 CA-074Me-treated animals). Note that Cath B activity in the liver was significantly higher than that in the cortex regardless of injury group. Additionally, ICP infusion of CA-074Me did not impact Cath B activity in the liver; however, CA-074Me infusion did significantly lower Cath B activity in the left and right cortex compared to DMSO controls. * *p* < 0.05 compared to injury type-matched vehicle control, # *p* < 0.05 compared to injury type-matched liver. Mean ± S.E.M.

**Figure 3 biomedicines-12-01612-f003:**
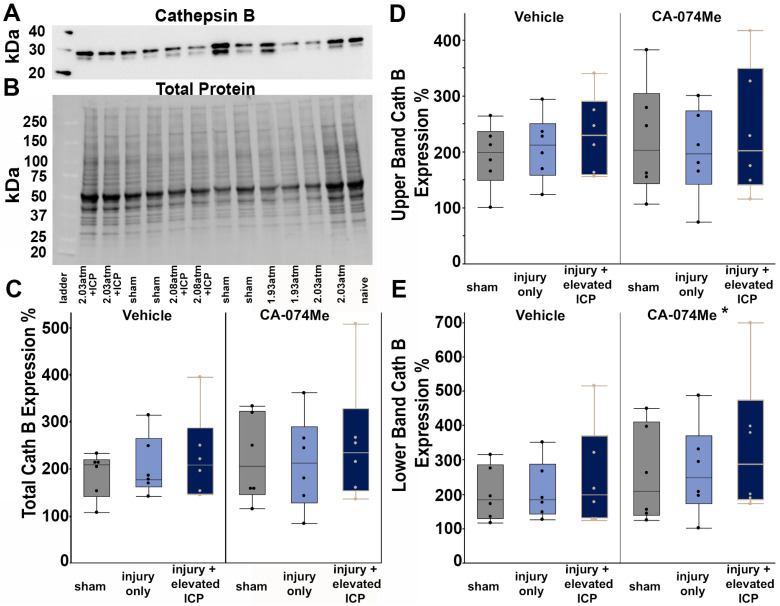
Cathepsin B (Cath B) protein levels were slightly increased in animals infused with the Cath B inhibitor CA-074Me. (**A**) Representative chemiluminescent blot image of the two mature Cath B bands at 24/27 kDa, which was normalized to (**B**) total protein. Box and whisker graphs depicting average (n = 6/group) quantities of (**C**) total Cath B protein, (**D**) the upper band of the Cath B doublet, and (**E**) the lower band of the Cath B doublet for sham-injured animals (grey boxes), animals sustaining a central fluid percussion injury (CFPI; light blue boxes), and animals sustaining both CFPI and secondary intracranial pressure (ICP) elevation (dark blue boxes) followed by a 2w intracerebroventricular infusion of either 10% DMSO (vehicle, left half of graphs) or the Cath B inhibitor, CA-074Me (right half of graphs). Amount of Cath B for each case was calculated as a percentage compared to a consistent naïve control that was run on all membranes. While there were no significant differences found for total Cath B protein or the upper band of the observed doublet, there was a significant increase in Cath B levels of the lower band of the doublet in animals infused with CA-074Me compared to their vehicle counterparts. * *p* < 0.05 compared to vehicle. Mean ± S.E.M.

**Figure 4 biomedicines-12-01612-f004:**
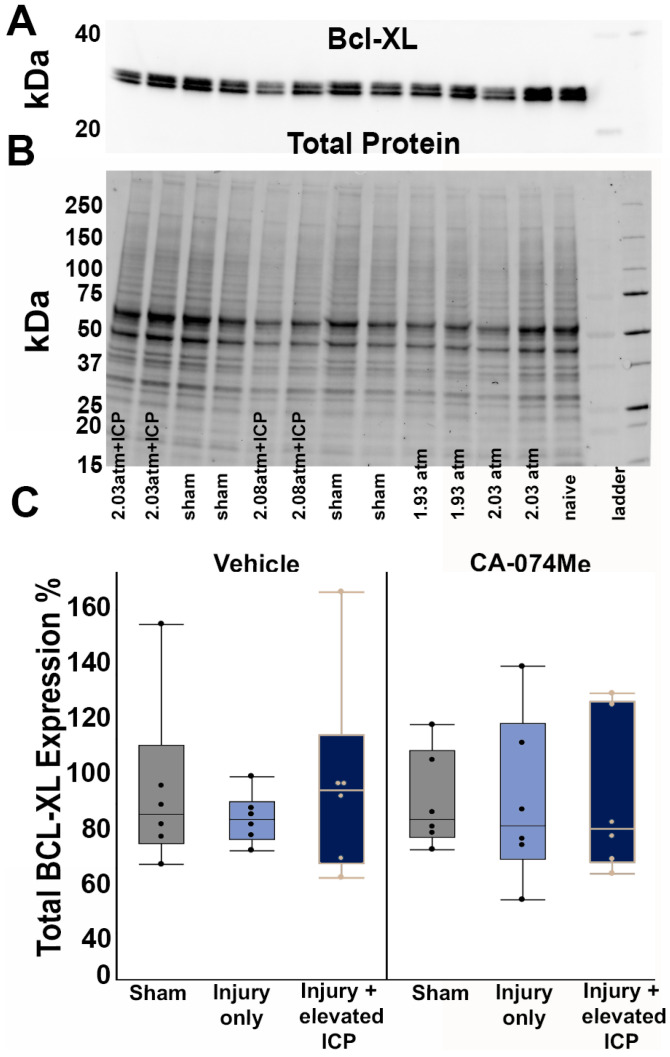
Protein levels of Bcl-XL were unchanged regardless of injury or infusion group. (**A**) Representative chemiluminescent blot image of Bcl-XL at 30 kDa, which was normalized to (**B**) total protein. (**C**) Box and whisker graph depicting average (n = 6/group) quantities of Bcl-XL protein for sham-injured animals (grey boxes), animals sustaining a central fluid percusion injury (CFPI; (light blue boxes), and animals sustaining both CFPI and secondary intracranial pressure (ICP) elevation (dark blue boxes) followed by a 2w ICF infusion of either 10% DMSO (vehicle, left half of graph) or the Cath B inhibitor, CA-074Me (right half of graph). Amount of Bcl-XL for each case was calculated as a percentage compared to a consistent naïve control that was run on all membranes. Mean ± S.E.M.

**Figure 5 biomedicines-12-01612-f005:**
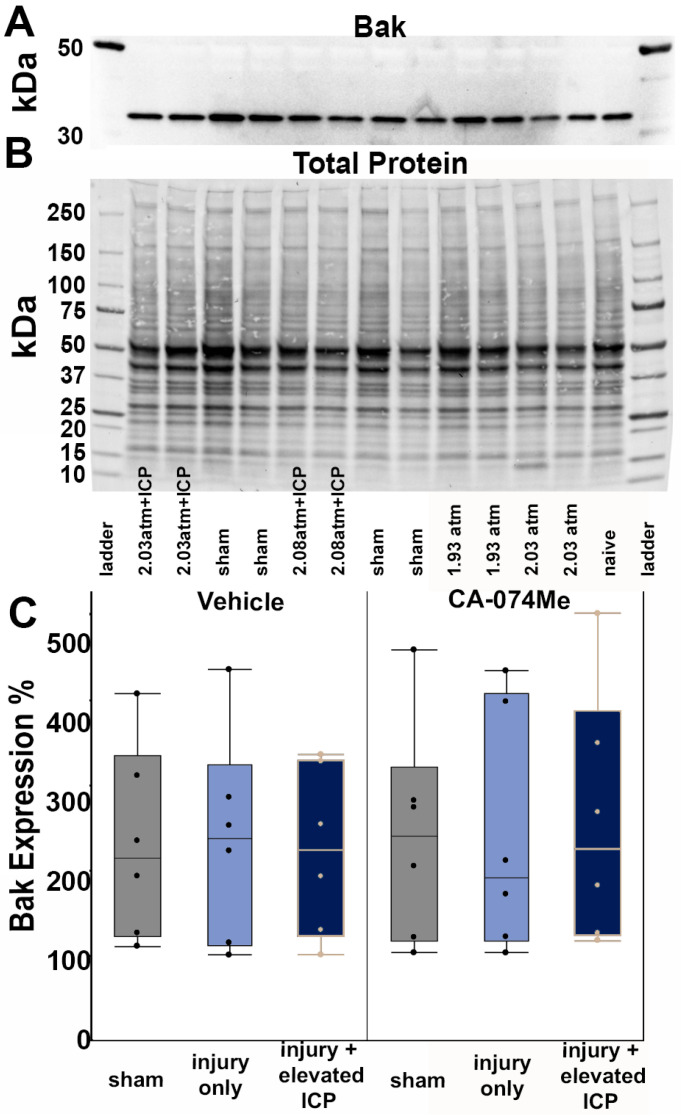
Protein quantification of BAK revealed no differences in the protein quantity regardless of group. (**A**) Representative chemiluminescent blot image of BAK at 25 kDa, which was normalized to (**B**) total protein. (**C**) Box and whisker graph depicting average (n = 6/group) quantities of BAK protein for sham-injured animals (grey boxes), animals sustaining a central fluid percussion injury (CFPI; light blue boxes), and animals sustaining both CFPI and secondary intracranial pressure (ICP) elevation (dark blue boxes) followed by a 2w ICF infusion of either 10% DMSO (vehicle, left half of graph) or the Cath B inhibitor, CA-074Me (right half of graph). Amount of BAK for each case was calculated as a percentage compared to a consistent naïve control that was run on all membranes. Mean ± S.E.M.

**Figure 6 biomedicines-12-01612-f006:**
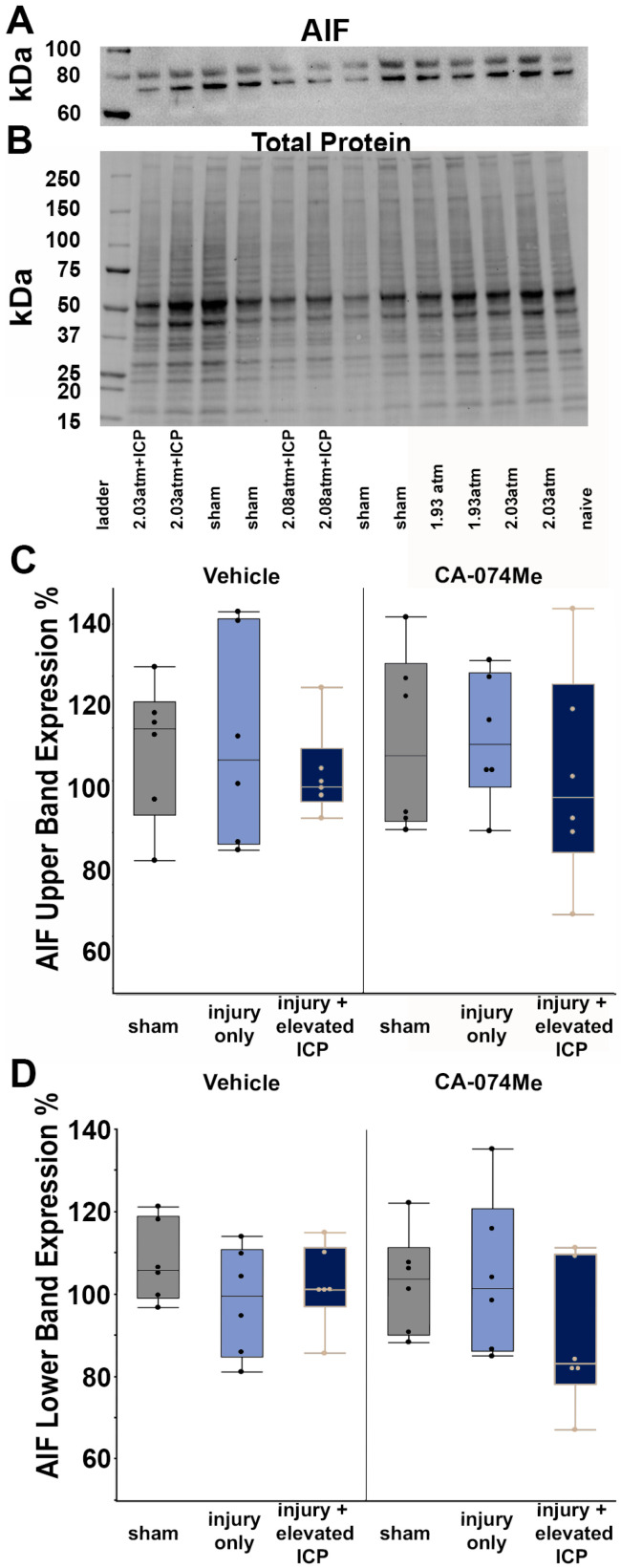
Protein quantification of AIF revealed no differences in the protein quantity regardless of group. (**A**) Representative chemiluminescent blot image of the two observed AIF bands at ~67/72 kDa, which was normalized to (**B**) total protein. Box and whisker graphs depicting average (n = 6/group) quantities of (**C**) the upper band of the AIF doublet and (**D**) the lower band of the AIF doublet for sham-injured animals (grey boxes), animals sustaining a central fluid percussion injury (CFPI; light blue boxes), and animals sustaining both CFPI and secondary intracranial pressure (ICP) elevation (dark blue boxes) followed by a 2w intracerebroventricular infusion of either 10% DMSO (vehicle, left half of graphs) or the Cath B inhibitor, CA-074Me (right half of graphs). Amount of AIF for each case was calculated as a percentage compared to a consistent naïve control that was run on all membranes. Mean ± S.E.M.

**Figure 7 biomedicines-12-01612-f007:**
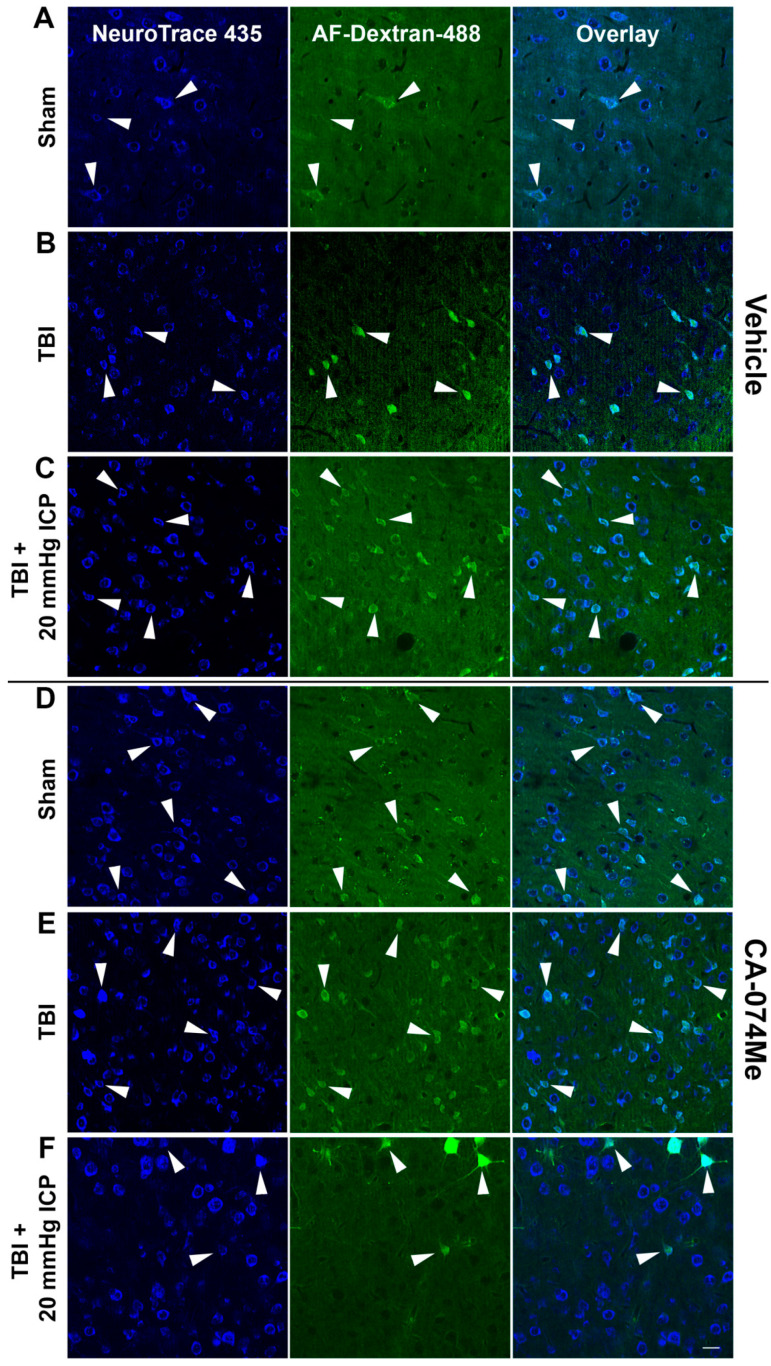
Representative fluorescent micrographs of membrane disruption in sham-injured animals (sham, (**A**,**D**)), animals sustaining a traumatic brain injury (TBI, (**B**,**E**)), and animals sustaining a TBI followed by secondary intracranial pressure (ICP) elevation (TBI+ 20 mmHg ICP elevation, (**C**,**F**)) paired with 2w of intracerebroventricular infusion of 10% DMSO (vehicle (**A**–**C**)) or the Cathepsin B inhibitor, CA-074Me (**D**–**F**). The left panel in blue depicts NeuroTrace Nissl-stained cells and the middle panel in green depicts cells containing a cell-impermeable Alexa-Fluor-488-tagged dextran (AF-dextran-488). The right panel is the overlay of the NeuroTrace and membrane-disrupted dextran images. The arrow heads indicate representative membrane-disrupted neurons. Scale bar 20 μm.

**Figure 8 biomedicines-12-01612-f008:**
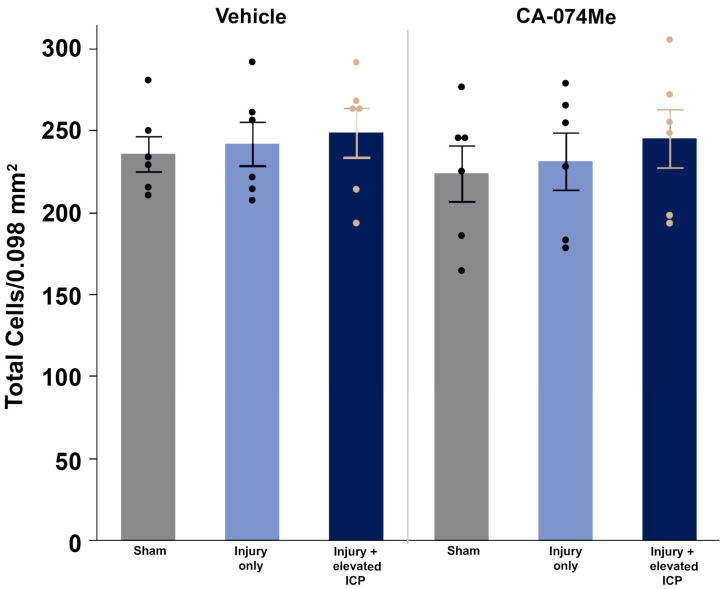
The total number of cells in the lateral neocortex layers V and VI is unaffected across injury and infusion groups. Bar graph depicting the average number of fluorescent NeuroTrace Nissl-stained neurons in sham-injured animals (grey boxes), animals sustaining a CFPI (light blue boxes), and animals sustaining both CFPI and secondary ICP elevation (dark blue boxes) followed by a 2w ICV infusion of either 10% DMSO (vehicle, left half of graphs) or the Cath B inhibitor, CA-074Me (right half of graphs). The mean number of cells was quantified per unit area (0.098 mm^2^) and averaged for each animal (n = 6/group). Mean ± S.E.M.

**Figure 9 biomedicines-12-01612-f009:**
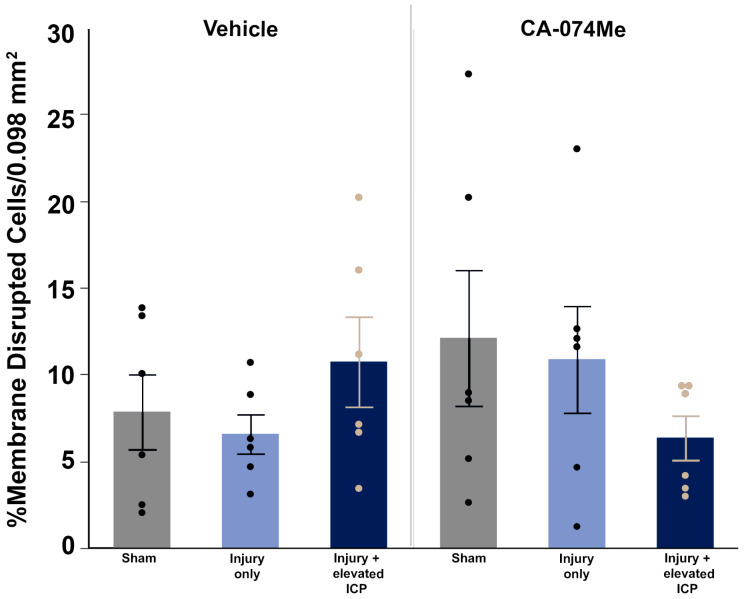
There were no significant changes in membrane disruption regardless of injury or Cathepsin B inhibitor infusion. Bar graph highlighting a consistently low average (n = 6/group) percentage of membrane-disrupted neurons in sham-injured animals (grey boxes), animals sustaining a central fluid percussion injury (CFPI; light blue boxes), and animals sustaining both CFPI and secondary intracranial pressure (ICP) elevation (dark blue boxes) followed by a 2w ICV infusion of either 10% DMSO (vehicle, left half of graph) or the Cathepsin B inhibitor CA-074Me (right half of graph). Mean ± S.E.M.

**Figure 10 biomedicines-12-01612-f010:**
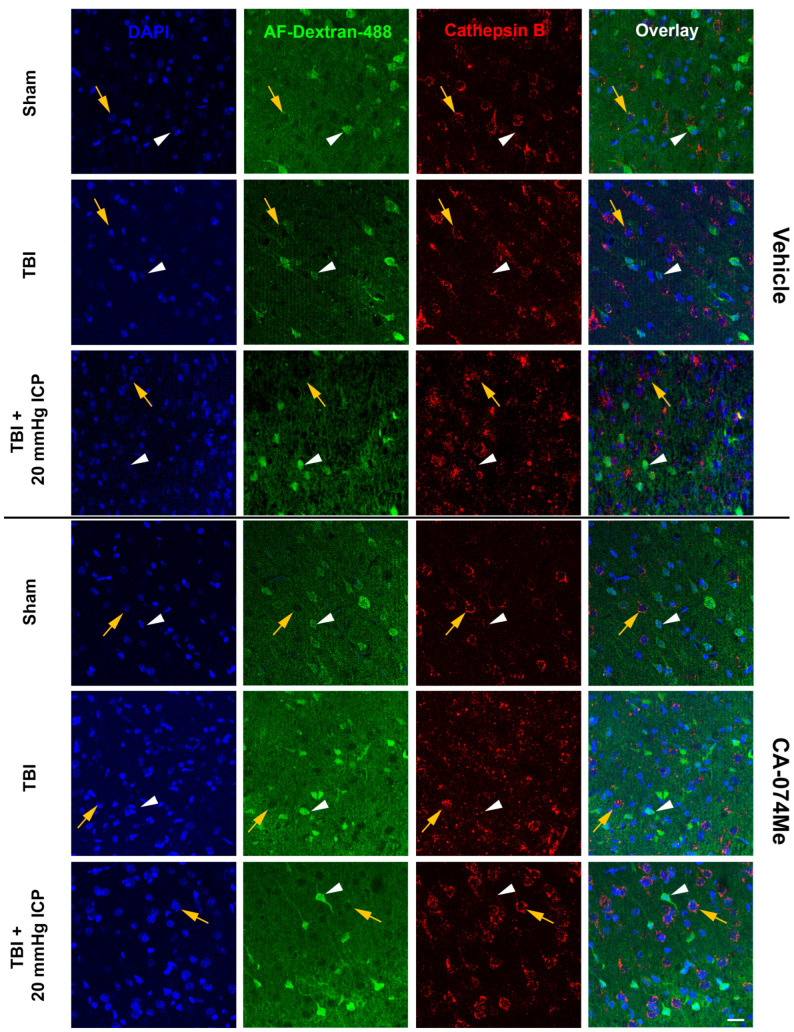
Representative fluorescent micrographs of cathepsin B (Cath B) localization in membrane-disrupted and non-disrupted neurons within the lateral neocortex layers V and VI in sham-injured animals, animals sustaining a traumatic brain injury (TBI), and animals sustaining a TBI followed by secondary intracranial pressure (ICP) elevation (TBI + 20 mmHg ICP elevation) paired with 2w of intracerebroventricular infusion of 10% DMSO (vehicle) or the Cath B inhibitor CA-074Me. The left-most panel contains DAPI-labeled nuclei (blue). Neurons were identified as non-disrupted (yellow arrows) or membrane-disrupted (white arrowheads) based upon uptake of cell-impermeable Alexa-Fluor-488-tagged 10 kDa dextran (AF-dextran-488; second panel in green), which diffused throughout the parenchyma. Immunolabeling for Cath B (third panel in red) allowed investigation of Cath B localization inside lysosomal puncta or outside lysosomes. The right-most panel is an overlay of the single channel images. Scale bar 20 μm.

**Figure 11 biomedicines-12-01612-f011:**
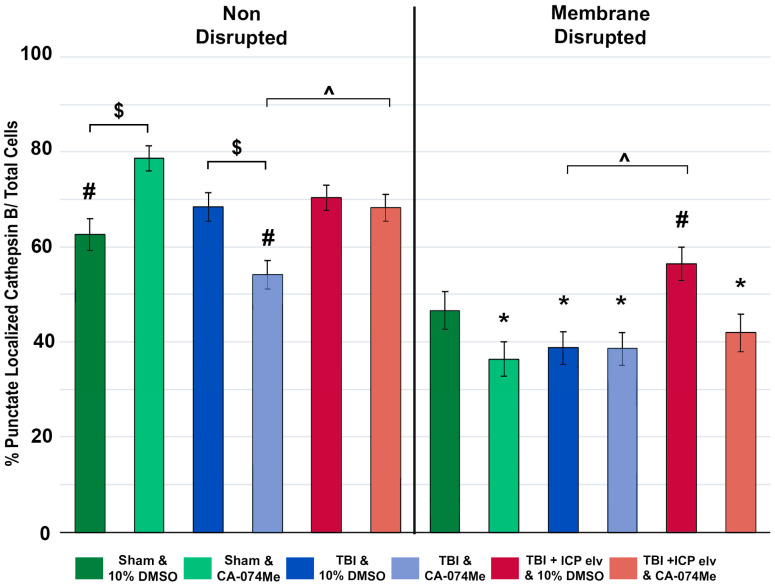
Cathepsin B (Cath B) inhibition with CA-074Me impacted the distribution of Cath B within lysosomal puncta. Bar graph depicting the percent of neurons within the non-disrupted (left half of graph; sham DMSO n = 214 neurons [dark green bars], sham CA-074Me n = 248 [light green bars], traumatic brain injury (TBI) DMSO n = 238 neurons [dark blue bars], TBI CA-074Me n = 275 neurons [light blue bars], TBI + intracranial pressure (ICP) DMSO n = 226 neurons [dark pink bars], and TBI + ICP CA-074Me n = 351 neurons [light pink bars]) and membrane-disrupted (right half of graph; sham DMSO n = 159 neurons [dark green bars], sham CA-074Me n = 176 neurons [light green bars], TBI DMSO n = 204 neurons [dark blue bars], TBI CA-074Me n = 205 neurons [light blue bars], TBI + ICP DMSO n = 153 neurons [dark pink bars], TBI + ICP CA-074Me n = 199 neurons [light pink bars]) populations that exhibited punctate localization of Cath B. Note that Cath B localization within puncta was impacted by inhibitor infusion, injury group, and membrane disruption status of the neurons analyzed. * *p* < 0.05 compared to non-disrupted counterpart, # *p* < 0.05 compared to sham and CA-074Me, $ *p* < 0.05 compared to vehicle-treated counterpart, ^ *p* < 0.05 compared to TBI-only infusion group counterpart. Mean ± S.E.M.

**Figure 12 biomedicines-12-01612-f012:**
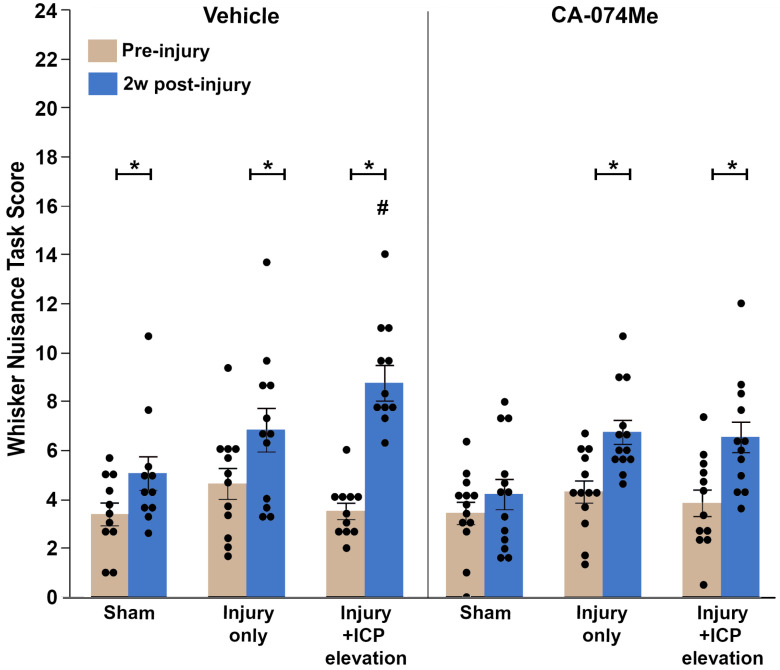
Somatosensory sensitivity was impacted by injury, particularly with secondary ICP elevation, and further impacted by cathepsin B inhibition with CA-074Me. Bar graph depicting the average whisker nuisance task (WNT) score pre-injury (tan bars) and at 2w post-injury or sham (blue bars). Sham animals infused with 10% DMSO (n = 11) had slightly higher WNT scores post-injury compared to sham animals infused with CA-074Me (n = 13). Animals sustaining traumatic brain injury (TBI) infused with 10% DMSO (n = 12) demonstrated similar post-injury WNT scores as TBI animals infused with CA-074Me (n = 13). Injured animals with secondary intracranial pressure (ICP) elevations (TBI + 20 mmHg ICP) infused with CA-074Me (n = 12) had lower WNT scores compared to injured and ICP-elevated animals infused with 10% DMSO (n = 11), resulting in an overall reduction in post-injury WNT score in animals infused with CA-074Me. * *p* < 0.05 compared pre-injury WNT score for that group, # *p* < 0.05 compared to sham 10% DMSO. Mean ± S.E.M.

## Data Availability

The raw data associated with this manuscript are appended as Appendix A. In addition, these data, along with the protocols, will be submitted to the Open Science Framework for open data access.

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
