# Peer review of "The Effects of Cathepsin B Inhibition in the Face of Diffuse Traumatic Brain Injury and Secondary Intracranial Pressure Elevation"

_biomedicines, 2024, doi:10.3390/biomedicines12071612_

Round 1

Reviewer 1 Report

Comments and Suggestions for Authors

In this manuscript, the authors explored the effect of Cath B inhibition in TBI and secondary ICP. The authors claimed this study demonstrated that Cath B is not a direct driver of membrane disruption, however, administration of CA-074Me alters Cath B localization and reduced hypersensitivity, emphasizing Cath B being an important component in late secondary pathologies. But in my opinion this manuscript this article did not produce any substantial results and cannot be published as a formal article for reference. In addition, I found some questions in the article and hope the author can answer them:

1.     The logic of the article is a little unclear, and a lot of space is used to briefly describe the negative results obtained.

2.     The authors used 0.9% saline to dilute CA-074 contained DMSO solution, but I think this method will cause the drug to precipitate and affect the drug's effect. Have the authors considered using other co-solvents to dissolve the drug, such as PEG300 and Tween 80?

3.     In WB result, how did the author normalized the target protein to total protein? why didn't the authors use internal reference proteins, such as GAPDH or actin?

Author Response

Comment 1: The logic of the article is a little unclear, and a lot of space is used to briefly describe the negative results obtained.

Response 1: We apologize for the lack of clarity regarding the logic of the current study and have edited the introduction to better clarify this. Since the specific impact of Cath B inhibition following CFPI was unknown, the focus of this study was to evaluate the effects of continuous Cath B inhibition via intracerebroventricular infusion of the Cathepsin B inhibitor, CA-074Me, for 2w following injury with or without secondary elevation of ICP. While we do appreciate that most of our results demonstrate no significant impact of cathepsin B reduction in downstream protein expression and neuronal membrane disruption, we do still maintain that these results are valid and worth publication, especially considering the reproducibility crisis that highlighted the lack of published negative results as a potential driving factor.

Comment 2:  The authors used 0.9% saline to dilute CA-074 contained DMSO solution, but I think this method will cause the drug to precipitate and affect the drug's effect. Have the authors considered using other co-solvents to dissolve the drug, such as PEG300 and Tween 80?

Response 2: The reviewer is correct that at higher dilutions with 0.9% saline (<10% DMSO) or when the compound was frozen in any condition other than in 100% DMSO, the CA-074Me did precipitate. As we determined the concentration and protocols for CA-074Me processing and infusion, any time the compound did precipitate out of solution it was visually apparent. To avoid the confound of precipitation, we froze aliquots of CA-074Me in 100% DMSO at 100ug/ul concentrations and reconstituted each aliquot individually with 0.9% saline to a final concentration of 10% DMSO and 10ug/ul CA-074Me. This method prevented precipitation of the inhibitor. We further generated animals process for molecular studies alongside those used for histological studies to verify that the CA-074Me was inhibiting Cath B throughout our investigations. We have now included these details to the methods section (lines 103-105 and 115-118). 

Comment 3:  In WB result, how did the author normalized the target protein to total protein? why didn't the authors use internal reference proteins, such as GAPDH or actin?

Response 3: We appreciate that many western blot studies do use a reference protein, such as actin or GAPDH, for protein normalization. However, as traumatic brain injury impacts many processes and protein expressions, including some indications that actin (DOI:10.1523/JNEUROSCI.0408-12.2012) and GAPDH (DOI: 10.1016/j.ab.2023.115301) might be altered following TBI, we could not verify that any of the potential normalization proteins would adequately be representative of the total protein being loaded into each lane. Therefore, we opted to utilize total protein for loading normalization to avoid issues with potential changes in any normalization protein following TBI. We have added this to the methods section for clarity (lines 251-252).

Reviewer 2 Report

Comments and Suggestions for Authors

The presented study “The effects of cathepsin B inhibition in the face of diffuse traumatic brain injury and secondary intracranial pressure elevation” is certainly relevant, as it is aimed at studying the important problem of traumatic brain injury. The study was well implemented, primary data and a good quality Western blot were presented. However there are some notes:

1. Data on the activity of DMSO aimed at the preservation of neurons do not seem to be very proven since they are based on photographs obtained during immunofluorescent staining (Fig. 9). However, the images are not of the best quality; there is a strong background of green fluorescence. Perhaps it is worth improving the illustrative material somewhat.

2. In future studies, I would like to recommend using ethanol rather than DMSO to dissolve non-polar substances; it is a solvent with a smaller spectrum of undesirable activity.

3. This manuscript represents a large research work, so it is extremely important to present a diagram of the study conducted, indicating the timing of all procedures and research methods performed.

4. It is necessary to present in the manuscript aligned images obtained by the Western blot method for Bcl-xL and AIF (Fig. 3, 5).

5. If possible, Figure 9 should be improved and symbols (arrows) added similar to Figure 6.

Author Response

Comment 1: Data on the activity of DMSO aimed at the preservation of neurons do not seem to be very proven since they are based on photographs obtained during immunofluorescent staining (Fig. 9). However, the images are not of the best quality; there is a strong background of green fluorescence. Perhaps it is worth improving the illustrative material somewhat.

Response 1: We have now updated Figure 9 (now Figure 10) to demonstrate the relocalization of cathepsin B more clearly. We have also included the description of the Alexa-488 dextran, which permeates through the parenchyma as it diffuses from the lateral ventricle leading to the high level of green dextran visible within the parenchyma to the figure legend.

Comment 2: In future studies, I would like to recommend using ethanol rather than DMSO to dissolve non-polar substances; it is a solvent with a smaller spectrum of undesirable activity.

Response 2: Thank you for the suggestion. We will not be using DMSO and will rather switch to using ethanol to dissolve the CA-074Me in future studies.  

Comment 3: This manuscript represents a large research work, so it is extremely important to present a diagram of the study conducted, indicating the timing of all procedures and research methods performed.

Response 3: We have now included a schematic representation of the experimental design as Figure 1 of the manuscript.

Comment 4. It is necessary to present in the manuscript aligned images obtained by the Western blot method for Bcl-xL and AIF (Fig. 3, 5).

Response 4: We have now more fully aligned the total protein images with the chemiluminescent images in Figure 3 (now Figure 4) and Figure 5 (now Figure 6). 

Comment 5: If possible, Figure 9 should be improved and symbols (arrows) added similar to Figure 6.

Response 5: We have updated Figure 9 (now Figure 10) for clarity and included arrows to specifically demonstrate the localization changes in cathepsin B within the membrane disrupted (white arrowheads) and non-disrupted (yellow arrows) neuronal populations that is quantified in figure 10.

Reviewer 3 Report

Comments and Suggestions for Authors

In current study the somatosensory changes, Cathepsin B (Cath B) activity, protein levels of Cath B and Cath B binding partners AIF, Bcl-XL, and Bak, as well as cell loss, membrane disruption, and Cath B localization were assessed in adult male Sprague-Dawley rats subjected to 2 weeks of Cath B inhibition following traumatic brain injury (TBI). Presented results indicate that membrane disruption is not directly caused by Cath B; although, moderated hypersensitivity and changes in Cath B localization might be the consequence of Cath B inhibitor administration. Overall, this highlights Cath B's significance in late secondary pathologies. Although the study is of sufficient significance and originality, there are several issues that need to be addressed:

1.         In section Abstract CA-074Me should be defined.

2.         In the section Materials and methods:

-           a significant portion of this section needs to be paraphrased, as the iThenticate report indicates its high percent match,

-           the additional behavioral tests would strengthen the presented results,

-           beyond alterations in the levels and localization of the examined molecules, do their gene expressions, etc., also undergo changes? The information should be provided,

-           it is necessary to provide schematic representation of the experimental design, including the groups, timeline when each method was conducted, etc.,

-           the reorganization of this entire section should be performed according to the timeline of when each method was conducted,

-           the number of permit for the use of animals should be provided,

-           the citation format for manufacturers should be standardized.

3.         The manuscript should be prepared in accordance with the requirements and standards of the journal Biomedicines (https://www.mdpi.com/journal/biomedicines/instructions).

Author Response

Comment 1:         in section Abstract CA-074Me should be defined.

Response 1: We have now included the full name, CA-074 methyl ester, and defined it as a cathepsin B inhibitor in the abstract. (lines 34-35)

Comment 2: A significant portion of the methods section needs to be paraphrased, as the iThenticate report indicates its high percent match,

Response 2: We apologize for the high percent match with our previous methods. As they are the same methods used previously in many instances they were stated consistently to our previous studies and those studies are cited. We have since, rephrased these methods to reduce the iThenticate high percent match with our previous papers, however, also understand that the methods must be stated in the clearest way possible to enhance replicability.

Comment 3:  the additional behavioral tests would strengthen the presented results,

Response 3: We agree that additional behavioral studies would be needed in future studies to more fully tease out the role of cathepsin B. We, however, did not do any additional behavioral assessments on these animals, but have added a statement to the discussion highlighting the need for further studies. (lines 548-549)

Comment 4:   beyond alterations in the levels and localization of the examined molecules, do their gene expressions, etc., also undergo changes? The information should be provided,

Response 4: We did not investigate transcriptome changes in this study, rather, focused on the final protein changes, as CA-074Me inhibits the cathepsin B protein. We have now added a statement specifically highlighting the lack of transcriptional data to the discussion. (lines 488-489)

Comment 5: it is necessary to provide schematic representation of the experimental design, including the groups, timeline when each method was conducted, etc.,

Response 5: We have now included a schematic representation of the study design that includes the timeline for each aspect of the study for both the animals used for histological assessments as well as those used for molecular assessments. This is included as Figure 1 and is cited throughout the methods.

Comment 6: the reorganization of the entire methods section should be performed according to the timeline of when each method was conducted,

Response 6: We have now reorganized the methods so that they more faithfully align with the experimental design timeline. We have also cited the experimental design timeline (Figure 1) throughout the methods for clarity.

Comment 7: the number of permit for the use of animals should be provided,

Response 7: We have now included the IACUC approval number to the methods section.

Comment 8:  the citation format for manufacturers should be standardized.

Response 8: We have updated the citations for manufacturers so that they are consistent throughout the manuscript highlighting the company, catalogue number, and headquarters location the first time a specific item is mentioned in the methods section. We also included RRIDs for all antibodies used.

Comment 9: The manuscript should be prepared in accordance with the requirements and standards of the journal Biomedicines (https://www.mdpi.com/journal/biomedicines/instructions).

Response 9: Thank you. The manuscript was prepared with the Biomedicines word template and includes all sections specified.

Round 2

Reviewer 1 Report

Comments and Suggestions for Authors

All the questions I raised have been solved. I think this article can be published.

Author Response

Thank you for your time and comments. 

Reviewer 3 Report

Comments and Suggestions for Authors

The authors have addressed most of the concerns and improved the quality of the manuscript. However, a detailed review of the manuscript is necessary, as several typographical errors have been noted.

Comments on the Quality of English Language

Minor editing of English language required

Author Response

Thank you for your time and comments. We have gone through the manuscripts and edited typographical errors.